# Molybdenum Cofactor Deficiency in Humans

**DOI:** 10.3390/molecules27206896

**Published:** 2022-10-14

**Authors:** Lena Johannes, Chun-Yu Fu, Günter Schwarz

**Affiliations:** Institute of Biochemistry, Department of Chemistry & Center for Molecular Medicine Cologne, University of Cologne, 50674 Cologne, Germany

**Keywords:** molybdenum cofactor deficiency, cyclic pyranopterin monophosphate, cPMP, Nulibry, molybdopterin, *MOCS1*, *MOCS2*, *MOCS3*, *GPHN*, sulfite oxidase

## Abstract

Molybdenum cofactor (Moco) deficiency (MoCD) is characterized by neonatal-onset myoclonic epileptic encephalopathy and dystonia with cerebral MRI changes similar to hypoxic–ischemic lesions. The molecular cause of the disease is the loss of sulfite oxidase (SOX) activity, one of four Moco-dependent enzymes in men. Accumulating toxic sulfite causes a secondary increase of metabolites such as S-sulfocysteine and thiosulfate as well as a decrease in cysteine and its oxidized form, cystine. Moco is synthesized by a three-step biosynthetic pathway that involves the gene products of *MOCS1*, *MOCS2, MOCS3,* and *GPHN*. Depending on which synthetic step is impaired, MoCD is classified as type A, B, or C. This distinction is relevant for patient management because the metabolic block in MoCD type A can be circumvented by administering cyclic pyranopterin monophosphate (cPMP). Substitution therapy with cPMP is highly effective in reducing sulfite toxicity and restoring biochemical homeostasis, while the clinical outcome critically depends on the degree of brain injury prior to the start of treatment. In the absence of a specific treatment for MoCD type B/C and SOX deficiency, we summarize recent progress in our understanding of the underlying metabolic changes in cysteine homeostasis and propose novel therapeutic interventions to circumvent those pathological changes.

## 1. Introduction

Four molybdenum (Mo)-enzymes are known in humans, each catalyzing either catabolic or detoxifying reactions [1]. They all share an identical type of molybdenum cofactor (Moco) that is strictly required for the activity of all these enzymes. Sulfite oxidase (SOX) is considered, by far, the most critical enzyme for human health as it catalyzes the terminal step in oxidative cysteine catabolism, the oxidation of sulfite to sulfate. SOX is localized in the intermembrane space of mitochondria and links sulfite oxidation to the reduction of cytochrome c. There are two cytosolic Mo-enzymes, xanthine oxidoreductase and aldehyde oxidase, which are closely related Mo–iron–flavin enzymes catalyzing the hydroxylation reactions of purines and various heterocyclic substrates [1]. The mitochondrial amidoxime-reducing component (mARC) has been identified as a fourth vertebrate Mo-enzyme [2] and reduces N-hydroxylated prodrugs as well as metabolites such as N-hydroxy-L-arginine.

Patients with isolated deficiencies in SOX or xanthine oxidoreductase are well known in the field [3], while deficiencies in aldehyde oxidase and the amidoxime-reducing component have not been reported yet. The largest fraction of patients with deficiencies in those enzymes is represented by defects in the biosynthesis of the molybdenum cofactor, a highly conserved multistep pathway that involves multiple gene products [4]. Moco biosynthesis is divided into three major steps based on two intermediates, cyclic pyranopterin monophosphate (cPMP) [5,6] and molybdopterin (MPT) [7], that were identified decades ago (Figure 1).

The first step in Moco biosynthesis, the conversion of GTP into cPMP, is catalyzed by two proteins encoded by the *MOCS1* locus [8]. *MOCS1* expression undergoes alternative splicing, yielding two different types of transcripts, one bicistronic and one monocistronic mRNA [9]. The bicistronic transcript encodes for two open reading frames, MOCS1A and MOCS1B [10], of which only the first is translated into a functional MOCS1A protein. MOCS1A catalyzes the S-adenosyl-methionine and [4Fe-4S] cluster-dependent radical conversion of GTP to 3′ 8-cyclo-7,8 dihydro GTP (Figure 1). The second monocistronic transcript produces a MOCS1AB fusion protein that is proteolytically cleaved to release a MOCS1B protein [11] that converts 3′ 8-cyclo-7,8 dihydro GTP to cPMP [12]. We have shown that, in addition, alternative splicing will produce protein variants that target either mitochondria or the cytosol [11].

The second step in Moco biosynthesis is catalyzed by MPT synthase, which converts cPMP into molybdopterin (MPT) and is encoded by the bicistronic *MOCS2* gene producing both subunits (MOCS2A and MOCS2B) by a ribosomal leaky scanning mechanism [13]. Heterotetrameric MPT synthase [14] transfers stepwise two sulfides from the thiocarboxylated C-terminus of MOCS2A to cPMP, thus giving rise to a mono-thiolated pterin intermediate [15] (Figure 1). *MOCS3* encodes for the Moco sulfurase [16], which is required for the ATP-dependent thiolation of MOCS2A. 

The third and final step in Moco synthesis involves the synthesis of MPT-AMP and the subsequent molybdate-dependent hydrolysis of MPT-AMP [17,18,19] (Figure 1), releasing Moco. Both reactions are dependent on the *GPHN* gene, which encodes for a multi-domain cytosolic protein named gephyrin, composed of an N-terminal G-domain (GPHN-G), a central domain, and a C-terminal E-domain (GPHN-E). Besides Moco biosynthesis, gephyrin protein functions as a cytosolic membrane-associated receptor-clustering protein, being essential for the formation of inhibitory synapses [20]. 

## 2. Clinical Presentation of Molybdenum Cofactor Deficient Patients

Depending on which step in Moco synthesis is impaired, MoCD is classified into three types (A, B, and C), with MoCD type A and B representing the vast majority of cases; MoCD type C is extremely rare due to a very severe presentation (see below). 

Biochemically, all three forms of MoCD are indistinguishable and highly similar to isolated SOX deficiency (iSOD). The latter has been reported in more than 50 cases [21], whereas over 200 cases of MoCD have been described so far [22], and many more are known to metabolic clinicians. The diagnostic hallmarks of MoCD and iSOD are the accumulation of sulfite, S-sulfocysteine (SSC), thiosulfate, and taurine, accompanied by a reduction in cystine/cysteine as well as homocysteine. Furthermore, a reduction in pyridoxal 5′-phosphate (PLP) has been reported. In addition, as a matter of differential diagnosis, in MoCD patients—but not in iSOD patients—uric acid levels are strongly and progressively declined, while xanthine and hypoxanthine, substrates of another Mo-enzyme, xanthine oxidoreductase, are increased. In addition, urothione, the catabolic end product of Moco, can also be used to differentiate between MoCD and iSOD patients. Surprisingly, urothione synthesis requires thiopurine methyl transferase, a well-known drug-metabolizing enzyme [23].

Typical newborn MoCD patients present with a broad spectrum of clinical severity, with the vast majority being severely affected from the neonatal age. Most patients initially appear healthy, while some display minor dysmorphic facial features and may have solitary cerebral parenchymal cysts and hypoplastic pons and cerebellums. 

The most common presentation of severe, classical MoCD, as first described in 1978 [24], is of early myoclonic encephalopathy, often starting within hours to days after birth, with poor feeding, irritability, and a distressed facial expression, and quickly progressing to myoclonic seizures, decreased consciousness, and apnea. The electroencephalogram (EEG) can initially be normal, advancing to a generalized burst suppression pattern. Within the first two weeks, children may regain alertness but show persistent hyperexcitability, frequent myoclonus, tonic spasms and focal seizures with eye deviation, and facial flushing. Seizures are often refractive to anticonvulsants. Infants can display dystonic episodes with prominent limb hypertonia and opisthotonus. A proportion of children develop lens dislocation during infancy and nephrolithiasis has also been reported. Mortality is high due to intercurrent lower respiratory tract infections and seizures, with a reported median survival of 2.4 [22] or 3 years [25].

Neuroimaging usually demonstrates severe abnormalities. An early stage of generalized edema is quickly followed by features mimicking severe generalized hypoxic–ischemic encephalopathy [26], which evolves within a few weeks to a characteristic appearance, including cortical atrophy and loss of white matter with cyst formation, hypoplastic corpus callosum, abnormal basal ganglia, hydrocephalus ex vacuo, dilated ventricles, cerebellar and brainstem hypoplasia, and mega cisterna magna [26,27,28].

Increasingly, cases with symptom onset later in childhood and attenuated severity have been described. Occasionally, children present merely with dystonia and speech delay [29,30], and cranial imaging may only show changes to the basal ganglia or even appear normal [22,31]. Attenuated disease probably reflects a slightly higher residual activity of SOX and/or Moco synthesis, and the diagnosis is easily missed if specific diagnostic investigations are not undertaken. Secondary deterioration can occur with intercurrent illness.

## 3. Genetics of Moco Deficiency

### 3.1. MOCS1 Mutations Lead to cPMP Deficiency

The first and most complex step in Moco biosynthesis is catalyzed by MOCS1A and MOCS1B proteins, which catalyze the stepwise conversion of the purine GTP to cPMP (Figure 1). Functional aspects of MOCS1 proteins are derived from biochemical studies on their bacterial orthologues, MoaA and MoaC. MOCS1A belongs to the superfamily of radical S-adenosylmethionine (SAM) enzymes [32], being characterized by a [4Fe-4S] cluster, the radical SAM-cluster, which reductively cleaves SAM, producing an adenosyl radical by oxidizing the [4Fe-4S] cluster from +1 to +2 [33]. In addition, MOCS1A represents one of only eight (so far) classified radical SAM enzymes [32] harboring a second C-terminal auxiliary [4Fe-4S] cluster [34,35,36,37]. Both [4Fe-4S] clusters are coordinated by three cysteines, resulting in the high oxygen sensitivity of both clusters compared to [4Fe-4S] clusters coordinated by four cysteines. Following the formation of the adenosyl radical, the abstraction of an H-atom from the C3 of the GTP coordinated by the auxiliary [4Fe-4S] cluster of MOCS1A will lead to the cyclization of GTP, thus forming 3′,8-cylodihydro-GTP (3′,8-cH_2_-GTP) [38,39]. MOCS2B, a trimer of functional dimers, converts 3′,8-cH2-GTP to cPMP with the formation of cyclic phosphate as the thermodynamic driver for pterin ring formation [40].

The unique chemical nature of cPMP was first described in 2004, representing a unique geminal diol at the C1′ position and demonstrating the tetrahydropyranopterin nature of this first and rather stable intermediate of the pathway [6]. It is important to acknowledge this significant chemical nature of cPMP, given the observed reactivity of cPMP with the downstream MPT synthesis reaction being initiated at the C2′ position (see below).

The *MOCS1* gene consists of ten exons, of which exons 1 to 9 encode for MOCS1A and exon 10 for MOCS1B [41]. Interestingly, alternative splicing in exon 9 produces either monocistronic or bicistronic transcripts that lead to the expression of either MOCS1A or the fusion protein MOCS1AB, respectively. In addition, alternative splicing of exon 1 results in transcripts that lead to MOCS1A proteins targeted to mitochondria, while other variants remain in the cytosol [11]. In contrast, all expressed MOCS1AB protein variants are targeted to mitochondria due to an internal targeting motif localized upstream of the MOCS1B domain. Following the translocation of MOCS1AB to the mitochondrial matrix, proteolytic cleavage at position 432 of MOCS1AB results in the release of a catalytically active MOCS1B protein [11].

Mutations in *MOCS1* lead to MoCD type A, the most prevalent form of MoCD, affecting approximately 50–60% of all known patients to date [42,43]. The last and comprehensive review of all reported MOCS1 variants lists a total of 32 disease-causing variants, out of which 20 mutations represent loss of function variants leading to a complete lack of Moco and, therefore, a full penetrance of the disease. In addition, there are numerous missense variants reported that affect highly/invariant residues involved in iron–sulfur cluster binding or catalysis that are also expected to have a complete loss in activity. A recent review from 2021 by Misko et al. provides an additional reference to the novel and potentially disease-causing MOCS1 variants [43] that can be found in the Exome Aggregation Consortium and Genome Aggregation Database. We used a comprehensive bioinformatic approach to investigate the pathogenicity of naturally occurring MOCS1 variants and found multiple novel disease-causing variants that allow a first genetics-based prediction of the incidence of MoCD type A, being in the range of 1:200,000 to 1:500,000, without considering other cofounding factors such as regional consanguinity (Mayr et al. unpublished results). In general, and in agreement with the large number of patients diagnosed at a late stage with severe disease [25], MoCD and MoCD type A, in particular, are considered to be underdiagnosed. However, functional studies have also disclosed a misdiagnosed case [44] carrying a highly frequent c.1064T > C allele, leading to the missense variant I355T, which is fully active (Mayr et al. unpublished results). Reinspection of the original work additionally confirmed a hemizygous condition, excluding MoCD as the underlying cause of the reported encephalopathy.

Amongst other novel *MOCS1* variants reported in recent years [45,46,47], there was a recent, only mildly affected patient with a mutation that one would classify as loss of function. The underlying homozygous c.1338delG *MOCS1* mutation causes a frameshift (p.S442fs), with a premature termination of the MOCS1AB translation product at position 477 that suggests the lack of the entire MOCS1B domain. However, a comprehensive biochemical analysis demonstrated an unusual mechanism of translation re-initiation in the *MOCS1* transcript; this resulted in trace amounts of functional MOCS1B protein that were sufficient to partially protect the patient from the most severe symptoms of MoCD. In aggregate, 20 years of genetic studies on *MOCS1* have identified nearly 50 pathogenic variants, with most of them causing a complete loss of function, thus representing the largest group of MoCD patients.

### 3.2. Loss of MPT Synthesis Is Caused by Mutations in MOCS2

The second step of Moco synthesis involves the stepwise conversion of cPMP to MPT, resulting in the opening of the cyclic phosphate in cPMP and the insertion of the unique dithiolene required to chelate the molybdenum atom. Both sulfur atoms are derived from cysteine and are inserted one after the other, with the formation of a monosulfurated pterin species, as deduced from studies on the bacterial enzyme [15]. The overall reaction is catalyzed by MPT synthase, which represents a dimer of MOCS2A/B heterodimers, with MOCS2B mediating heterotetramer formation. Two active sites are formed at each MOCS2/B heterodimer, with the C-terminal thiocarboxylated tail of MOCS2A being inserted into MOCS2B, forming the binding site for cPMP. Following thiol transfer, MOCS2A dissociates from the MPT synthase complex and undergoes an ATP- and sulfide-dependent re-thiolation catalyzed by MOCS3 [48].

The *MOCS2* gene contains seven exons; exons 1–3 encode for 88 residues of the small subunit, MOCS2A, and exons 3–7 encode for the large subunit, MOCS2B. As a very rare situation in higher eukaryotes, the last 77 bps of exon 3 encode in two different open reading frames for two proteins, the C-terminal end of MOCS2A, harboring the catalytically essential double glycine motif, and the N-terminus of MOCS2B. Translation of both overlapping open reading frames is ensured by a ribosomal leaky scanning mechanism [49]. 

From the published case reports and review articles so far, we know that about 1/3 of all MoCD patients belong to MoCD type B, with the vast majority of them representing classical early onset patients. Here, we summarized 31 *MOCS2* mutations and 3 *MOCS3* mutations (Table 1). Among all *MOCS2* mutations, 11 of them are located on exon 1 or -2, encoding for MOCS2A; another six mutations reside in exon 3, affecting both MOCS2A and MOCS2B proteins, and another 14 variants were found to impact MOCS2B due to mutations in exons 4–7. Approximate 50% of the *MOCS2* mutations lead to either frameshifting or premature termination, causing a total loss of function of MPT synthase, while the other half represent missense mutations affecting protein complex formation and/or catalysis.

MoCD type B patients are usually reported with severe phenotypes due to a large number of severe mutations. However, the relation between the MPT synthase activity and the different missense mutations remains largely unknown. Only six MOCS2 variants have been characterized in three studies [62,67]. In the first study, the missense variant MOCS2A-S15R and truncated variant MOCS2B-A150∆ did not yield any stable protein for further analysis. In the size exclusion chromatography, the MOCS2A-V7F variant was impaired in a heterotetramer formation with MOCS2B, yet the MOCS2B-E168K variant formed a complex with MOCS2A, which retained 50% of WT MPT synthase activity, whereas the MOCS2A-V7F/MOCS2B complex showed 10% residual activity. Notably, the patient who carried MOCS2A-V7F was reported as a mild case, while the patient who carried the MOCS2B-E168K variant was reported as a severe case despite the much higher residual activity recorded in vitro [67]. In the second study, the MOCS2B-S140F variant yielded much lower protein than WT and revealed an alteration in protein folding. The change in the content of helical structures might influence either the oligomerization between MOCS2B protomers or the interaction with MOCS2A subunits; this was further confirmed by binding studies using isothermal titration calorimetry. As a consequence, MOCS2B-S140F was able to form MPT but at a much lower rate than WT MOCS2B [15]. In the latest study, the variant MOCS2B-L158_K159del was characterized. The structure analysis indicated that residues Leu158 and Lys159 are located at the end of the last helix and are involved in the binding of MPT synthase small subunits. Thus, the heterodimer and active complex formation were impaired in this variant [68]. In conclusion, these studies demonstrate a link between residual Moco synthesis activity and delayed onset and/or a milder course of the disease.

### 3.3. Patients with GPHN Mutations Show a Broad Spectrum of Neurological Disorders

The final step in human Moco biosynthesis is catalyzed by the multi-domain protein gephyrin, also representing a domain fusion of individual enzymatic functions of individual bacterial enzymes. Hereby, the G- and E-domains of gephyrin catalyze the adenylylation of MPT and the insertion of molybdate, suggesting product–substrate channeling as the underlying driving force for domain fusion [13,69]. Crystal structures of both catalytic domains have been determined in the past [70,71], while a full-length structure of the gephyrin is still lacking due to its high conformational flexibility [72].

As the name depicts, *GPHN* does not follow the classical *MOCS* gene nomenclature of all other genes involved in Moco synthesis due to its initial identification as a glycine receptor-associated protein [73]. At that time, the reported binding to microtubules was the main driver of naming the protein according to a proposed bridging function. Today, it is known that, similar to other postsynaptic scaffolding proteins, gephyrin fulfills a major structural and signaling function at inhibitory synapses in the central nervous system [20]. Gephyrin binds to the large intracellular domains of various subunits of glycine and GABA type A receptors. The high-resolution crystal structure of the dimeric gephyrin E-domain in complex with a peptide derived from the beta-subunit of the glycine receptor discloses two independent, active sites on the E-domain, one involved in Moco synthesis, the other binding the receptor in a key and lock mechanism at the interface of two dimers [70]. 

Vertebrate gephyrin is encoded by the *GPHN* gene, composed of up to 40 exons, with a varying number of exons undergoing alternative splicing [74]. Although the number of exons and alternative splice products reported in humans is far lower [75] than recently reported in mice, it is expected that the highly mosaic gene expression also holds true in humans and reflects the complex regulation of gephyrin function in the central nervous system. In the liver, the majority of gephyrin is encoded by a splice variant harboring the so-called C3-cassette, suggesting that this variant is primarily involved in gephyrin’s metabolic function.

In contrast to MoCD types A and B, the number of identified MoCD patients with mutations in *GPHN* is extremely low. To date, only two cases have been reported [42,76], with mutations in *GPHN* presenting a very severe disease course. This finding is most likely due to the dual function of gephyrin in Moco synthesis and neuroreceptor clustering and is mirrored by the phenotype of gephyrin-deficient mice that die on the first day of life [77]), while MOCS1- and MOCS2-deficient mice survive for 8–11 days [78,79]. Along these lines, numerous studies have identified heterozygous mutations and haploinsufficiency in the *GPHN* gene, causing a variety of neuropsychiatric disorders [80,81,82] underlying the fundamental function of gephyrin in the central nervous system.

## 4. Therapies to Treat MoCD

### 4.1. Disease Mechanisms

Most of the symptoms in MoCD patients are mirrored by iSOD, which is caused by mutations in the *SUOX* gene. SOX is one of four Mo-enzymes in mammals, and loss of SOX activity is considered the key enzyme contributing to the pathophysiology of MoCD. The hallmark of iSOD and MoCD is the accumulation of cytotoxic sulfite, which is highly reactive. For example, sulfite is able to reduce disulfide bridges in both small sulfur-containing molecules as well as in proteins. Several molecules were reported as sulfite scavengers, namely, cystine, oxidized glutathione (GSSG), homocystine, and cystamine; they form S-sulfonated species following their reaction with sulfite. The effects of these S-sulfonated species remain largely unknown except for S-sulfocysteine (SSC). SSC represents a structural analog of glutamate and has been proven to cause excitotoxicity in neurons by activating N-methyl-D-aspartate type glutamate (NMDA) receptors, causing calcium influx and downstream signaling such as calpain activation [83]. From many case reports describing iSOD and MoCD patients, it becomes obvious that disease progression is directly related to the severity of neurodegeneration. Consequently, targeting sulfite and its downstream targets (such as SSC) are the main strategies for future treatments (see below). 

Although sulfite and SSC are two well-known biomarkers that can be detected in MoCD and iSOD patients very shortly after birth, there are other changes in metabolites that hint at additional mechanisms that might contribute to MoCD and iSOD pathophysiology. First, cystine and cysteine are depleted due to SSC formation and increased renal excretion. Cysteine as the precursor of glutathione (GSH) and low cysteine/cystine levels could lead to reduced cellular GSH levels that could cause ferroptosis, a new form of non-apoptotic cell death (see Section 4.7). Second, MoCD patients were also reported to exhibit lower homocysteine levels that might be caused by the formation and increased excretion of S-sulfohomocysteine and/or reflect a secondary cause of cysteine depletion. Finally, a sulfite-dependent reduction of pyridoxal 5′-phosphate (PLP) in cerebrospinal fluid has been reported in MoCD patients [84]. The latter is considered a pro-excitatory effect of sulfite, which could also contribute to neurodegeneration in a similar way to pyridoxine-responsive encephalopathy.

Sulfite is the major downstream intermediate product of cysteine catabolism, which can be divided into two major pathways: Excess cysteine is converted via cysteine sulfinic acid to taurine (Tau) by the subsequent reactions catalyzed by cysteine dioxygenase (CDO) and cysteine sulfinic acid decarboxylase (Figure 2). Alternatively, aspartate:2-oxoglutarate aminotransferase (AST) converts cysteine sulfinic acid to β-sulfinyl pyruvate, which spontaneously decomposes to pyruvate and sulfite. Recently, we demonstrated that cytosolic AST is the major contributor to sulfite formation [85]. The second route of cysteine metabolism involves H_2_S, which is generated by three alternative routes involving, cystathionine γ-lyase (CSE), cystathionine β-synthase (CBS), and 3-mercaptopyruvate sulfurtransferase (MSPT). The latter enzyme also requires AST; however, here, the mitochondrial isoform of AST has been implicated as a major contributor [85].

The half-life of H_2_S is considered to be short as it rapidly metabolizes in mitochondria to either sulfite or thiosulfate; this involves three enzymes: sulfide quinone oxidoreductase (SQOR) catalyzes the H_2_S-dependent formation of persulfidate species such as glutathione-persulfide, which is further converted by protein sulfide oxidase (PDO) to sulfite. A third enzyme, thiosulfate sulfur transferase (TST), forms thiosulfate in the presence of sulfite and persulfides. 

### 4.2. Treatment of MoCD Type A (MOCS1) Patients with cPMP

To this day, an effective treatment strategy has only been established for MoCD type A. Following successful preclinical studies in a mouse model of MOCS1 deficiency [86], the first treatment of MoCD in a human patient was reported in 2010 [87]. In this patient, daily intravenous administration of purified *E. coli*-derived cPMP, the first intermediate of Moco biosynthesis, was initiated. 

Before treatment, the first patient presented with seizures, feeding abnormalities, and cerebral atrophy and displayed abnormally increased urinary levels of SSC, thiosulfate, xanthine, and sulfite. Gene analysis found a homozygous mutation in exon 10 of the *MOCS1* gene, leading to MoCD type A, in which the first step of Moco biosynthesis is disturbed. 

Following the diagnosis of MoCD type and legal approval by the local authorities, treatment of the index case was started with daily intravenous injections of 80 µg/kg cPMP at day 36 of life. After 12 and 35 days of treatment, the dosage was increased to 120 and 160 µg/kg per day, respectively, while after 75 days, the dosage was temporarily increased to 160 µg/kg twice per day. Within a few days after initiating cPMP therapy, urinary levels of SSC, thiosulfate, xanthine, and uric acid decreased drastically until the concentrations were stabilized at control levels and urine sulfite tests became negative. Treatment was tolerated well, without any visible side effects, and when examined by a pediatrician at 18 months, the child was clinically seizure-free and displayed no signs of progressive neurodegeneration. 

Since this first success story, more than 20 MoCD patients have benefited from cPMP substitution therapy. In a cohort study from 2015, eleven neonates were treated with intravenous cPMP, beginning between the ages of 0 to 68 days [88]. Disease biomarker levels were reduced almost back to normal within two days for all of them. Eight of the patients showed a significant improvement in their symptoms, with three of them even displaying near-normal development in the long term. Additionally, the treatment showed no serious adverse events after more than 6000 doses and was considered to be safe [88]. While in the early years, cPMP was enzymatically synthesized and purified from bacteria [86], chemical synthesis of cPMP was achieved in 2013 [89], leading to larger quantities and a more cost-effective production of cPMP. Following a clinical phase I safety study with synthetic cPMP, since 2014, patients treated with ‘recombinant’ cPMP were stepwise transferred to the new synthetic cPMP within the frame of a clinical phase II/III study (ORGN001, formerly ALXN1101; https://clinicaltrials.gov (accessed on 15 August 2022)). Finally, in 2021, chemically synthesized cPMP (named ‘Fosdenopterin’) was approved by the FDA and received marketing authorization as a Nulibry product [90]. It is administered via daily intravenous injections at a concentration of 400 µg/kg in pre-term infants and 550 µg/kg in term infants until it is increased to 900 µg/kg after the first three months of treatment or directly for children starting the therapy at the age of one year or later (Nulibry prescribing information [91]).

In MoCD types B and C, the later steps of Moco biosynthesis are affected, and, therefore, cPMP substitution is not applicable to those patients. MPT and Moco, the products of the second and third step of Moco biosynthesis, however, have a significantly shorter half-life than cPMP and are extremely sensitive to oxidation in aerobic environments [5], which, so far, have impeded their usage as a treatment for MoCD types B and C. Therefore, it is crucial to focus on the development of alternative therapeutic approaches that cover all known types of MoCD and iSOD.

### 4.3. Molybdate Treatment

In early studies, before knowing the underlying genetic and biochemical basis of MoCD, molybdate treatment was tested with the aim of increasing Moco formation [92]. While this was not successful, studies in fibroblasts derived from a *GPHN*-deficient patient demonstrated the possibility of a molybdate-dependent, partial restoration of Moco synthesis [76]. However, such an approach could only work in patients with *GPHN* missense mutations that do not affect the receptor-clustering function of gepyhrin, as reported for a single case harboring a point mutation affecting the E-domain [93]. 

Interestingly, the correct mitochondrial maturation of SOX is dependent on Moco [94], while the mitochondrial amidoxime-reducing component mARC1 is targeted to the outer mitochondrial membrane, independent of Moco [95]. Recently, two iSOD patients with impaired Moco binding have been reported [96,97], both of which exhibited an attenuated type of the disease, with milder symptoms. Biochemical characterization of those patients’ SOX variants (G362S, R366H) revealed an impairment in Moco coordination within the active site of SOX, resulting in strongly decreased mitochondrial maturation. Following the culture of patient fibroblasts in the presence of molybdate [96] or MOCS1 transgene expression [98], significantly increased SOX activity has been observed, thus demonstrating that the increase in Moco availability partially rescues a defect in Moco-dependent SOX maturation. Therefore, the molecular–genetic investigation of iSOD patients may identify future molybdate-responsive patients, opening the path to personalized medicine. 

### 4.4. Dietary Restriction

Until today, MoCD types B and C, as well as iSOD, are considered incurable diseases. In order to reduce the accumulation of toxic sulfite, dietary restriction of sulfur-containing amino acids was considered in the past for a number of patients [47,99,100]. The best results were obtained in mild cases of iSOD and MoCD type A, with dietary restriction of protein intake leading to significantly decreased urinary SSC and thiosulfate levels; in some cases, patients showed improved development without further signs of progressive neurodegeneration [47,100]. 

In a recent study by Abe et al. (2021), dietary restriction was used in a Japanese MoCD type A patient, as in Japan, cPMP treatment has not yet been officially approved or cPMP could not be obtained at the time of diagnosis. The patient carried a homozygous missense mutation at c.1510C > T (p.504R > W) in exon 10 of the *MOCS1* gene, which has not been reported before. It is located in the N-terminus of the catalytically essential MOCS1B domain, so it is expected to have a strong impact on the catalytic activity of MOCS1B protein. At the age of 16 months, dietary protein was restricted to 1.75 g/kg/day and further reduced in the following month to 1.4 g/kg/day. A further reduction to 1.25 g/kg/day was introduced until 42 months before the diet was finally continued with a protein content of 1.0 g/kg/day. After 4 months of dietary protein restriction, SSC values had decreased by about 50%, and urine sulfite tests became negative. Although the patient remained severely mentally defective, she developed without further regression and remained seizure-free [47]. 

Two patients with mild forms of iSOD were treated with dietary protein restriction combined with a particular decrease in sulfur-containing amino acids [100]. The mutations carried by both patients were not mentioned, thus not allowing any speculations on residual SOX activity. The first patient was diagnosed with iSOD late, at the age of 22 months, and, afterwards, began to receive 1.2 g/kg/day supplemented with an amino acid mixture not containing methionine, cystine, and taurine. Methionine intake was limited to 180 mg/day. After seven months, additional taurine (50 mg/kg), magnesium sulfate, and betaine treatment was started, and, two months later, protein intake was increased to 1.3 g/kg/day with 250 mg/day methionine. In this patient, treatment with cysteamine was also tested with the aim of quenching the accumulating sulfite (see below). The treatment, however, did not result in any clinical improvement, so it was stopped after nine months. At the age of 4 years and 5 months, the child presented with age-appropriate development and no signs of severe neurodegeneration. The second patient was diagnosed with a mild form of iSOD at the age of 10 months. At the beginning of treatment, dietary protein intake was severely decreased to 4.0 g/day (approx. 0.5 mg/kg/day), supplemented with an amino acid mixture that was free of methionine, cystine, and taurine. After two weeks, protein intake was adjusted to 1.0 g/kg/day with 125 mg methionine, which was further raised to 200 mg with increasing body weight and supplemented with taurine later on as well. Treatment resulted in very low urinary SSC and thiosulfate levels, which stabilized after the first two months of treatment. Two years and four months after the beginning of the therapy, the development of the child was progressing steadily, although the child remained timid and showed subnormal psychomotor skills.

The few cases reported above represent mild forms of iSOD and MoCD that benefited from dietary protein restriction. However, this therapeutic approach is not necessarily sufficient to rescue severe phenotypes of both diseases that present with a total loss of SOX activity. In contrast, a patient diagnosed with the classical, rapidly progressing onset of iSOD started a methionine- and cystine-free diet at the age of 12 days [101]. Additionally, he was treated with thiamine due to an observed sulfite-dependent depletion/reduction of thiamine as well as with dextromethorphan [101], a pharmacological blocker of NMDA receptors (see below). However, the patient displayed progressive microcephaly and presented with severe developmental delay, with no significant improvement upon treatment. For another severe case of iSOD, a decline in irritation but no neurological improvement upon methionine and cystine restriction has been reported [78]. These experiences suggest that dietary restriction of protein in general and sulfur-containing amino acids in particular may be an effective way of improving the phenotype of mild/attenuated cases of MoCD and iSOD but is not sufficient to prevent severe clinical courses. Moreover, the starting time of dietary restriction is crucial, as disease-related neurological damage is irreversible [87].

### 4.5. Targeting NMDA Receptors

Infant death in MoCD and iSOD patients is usually the result of severe neurodegeneration. As mentioned above, a major contributor to this pathology is SSC, causing an overexcitation of NMDA receptors, a mechanism that was suggested following the first experiments in rats almost 50 years ago [102]. In a recent study in 2017, SSC has been proven to be the main cause of neuronal cell death in in vitro experiments using primary cortical neurons [83]. The NMDA receptor antagonist MK801 was sufficient to rescue neurons from SSC-mediated cell death, strongly supporting the hypothesized mode of action of SSC. 

In the same study, a MoCD-like phenotype was induced in adult wild-type mice by treatment with the molybdenum antagonist tungstate [83]. Following cellular uptake, tungstate is considered to be incorporated in the same way as molybdenum to molybdopterin, thereby rendering it inactive and causing a loss of function of all Mo-enzymes. Following tungstate treatment, 4-week-old mice displayed a median decrease in SOX activity of 90% and developed a MoCD-like phenotype within three weeks, including weight loss and decreased neuromotor skills, as quantified in the Rotarod performance test [83]). Additionally, in contrast to *Mocs*1*^−/−^* [78] or *Mocs*2*^−^*^/*−*^ [79] mice, these animals showed decreased cell density and, thus, signs of neuronal degeneration in the cerebral cortex and the CA1 region of the hippocampus, hence resembling the phenotype of human MoCD and iSOD patients. Body weight and neuromotor skills were successfully rescued by intraperitoneal injection with 5 mg/kg of the NMDA receptor antagonist memantine twice per week; this makes NMDA receptor blockers in general and memantine in particular promising candidates for future therapeutic attempts. 

In humans, the approach of applying NMDA receptor antagonists has been studied only sporadically and, in most cases, in patients with severe disease progression. In a three-year-old boy with a mild form of MoCD and accompanying recurrent epilepsy, administration of the NMDA receptor antagonist dextromethorphan (12.5 mg/kg body weight) resulted in the disappearance of seizures, an improved EEG, and no major drug-related “side-effect” [103]. In contrast, in one of the aforementioned dietary approaches, dextromethorphan was administered in combination with a methionine- and cystine-free diet, starting at the age of three weeks [101]. The treatment, however, did not lead to a significant improvement of the patient’s health condition, which suggests that NMDA receptor blockage is again a convenient therapeutic approach in milder cases of MoCD and iSOD but is not sufficient to improve the phenotype in severely affected patients with manifested neurodegeneration.

### 4.6. Sulfite Scavenging

Given the largely overlapping pathophysiology in MoCD and iSOD, any therapeutic intervention that lowers the accumulation of toxic sulfite is considered to be beneficial for both disorders [104]. Therefore, chemical scavenging of excess sulfite to avoid its accumulation and the secondary formation of SSC with the accompanied depletion of cystine needs to be considered. Previous in vitro experiments revealed a 6-to-8-times faster reaction of sulfite with cystamine than with cystine [105], thus proposing cystamine as a potent sulfite scavenger preventing the formation of SSC. Interestingly, cysteamine, the reduced form of amino-thiol, represents an established treatment for nephropathic cystinosis and is used in concentrations of 0.2–0.6 mmol/kg [106]. As, in vivo, an equilibrium between the reduced and the oxidized form of any thiol-containing small molecular establishes rapidly, cysteamine and cystamine are expected to have a similar therapeutic effect. However, in vivo, the reduced thiol can not only form disulfide cystamine but also a mixed disulfide with one molecule of cysteine. Thus, it leads to further cysteine depletion, and 50% of the sulfite-derived reaction product will be, again, SSC. Therefore, the oxidized amino-cystamine might be favored for the treatment of sulfite toxicity. 

A single case of treatment with cysteamine in combination with dietary restriction [100] has been reported in the literature. The treatment had no effect on the patient’s overall health condition and was, therefore, stopped after 9 months. However, neither the concentration and frequency of cysteamine treatment nor biomarkers for sulfite intoxication were reported. In addition, it remained unclear to what extent irreversible brain damage might have masked any biochemical improvement. Given the sparse clinical experience and the expected underlying biochemical principles, sulfite scavengers still hold great potential as a possible therapeutic approach to treat sulfite toxicity in MoCD and iSOD.

### 4.7. Ferroptosis Inhibition 

Alteration in cysteine homeostasis in MoCD and iSOD hints at a novel form of regulated cell death—ferroptosis—that has been associated with reduced glutathione levels in mammals [107,108]. Ferroptosis is an iron-dependent form of non-apoptotic cell death that can be triggered by a depletion in cellular cysteine [109]. Cysteine is the rate-limiting component in the biogenesis of the antioxidant tripeptide glutathione as the other two required amino acids, glutamate and glycine, are found in significantly higher abundance within the cell [110]. A reduction in cysteine levels, therefore, causes a decrease in the biogenesis of glutathione, which is an essential cosubstrate in the detoxification of lipid peroxides catalyzed by glutathione peroxidase 4 (GPX4) [111]. Reduced levels of glutathione hamper this detoxification process, ultimately leading to increased oxidative stress and cell death. 

In 2012, cell death from a glutamate-induced blockade of system x_c_^-^-mediated cystine import in organotypic rat brain slices was first identified as an iron-dependent form of non-apoptotic cell death, which was then termed ferroptosis [112]. Since then, ferroptosis has been proposed to play a decisive role in the courses of multiple neurodegenerative diseases, including Alzheimer’s, Parkinson’s, and Huntington’s disease [113,114,115]. Cysteine enters the cell in its oxidized form, cystine, which is transported across the plasma membrane by the antiporter system x_c_^–^ in exchange for intracellular glutamate. In MoCD and iSOD, as a consequence of strongly elevated SSC formation derived from the reaction of cystine with sulfite, cysteine levels are strongly decreased [116,117]. In addition, due to its structural similarity to glutamate, an increase in extracellular SSC levels may inhibit the x_c_^-^ antiporter, thus further reducing the import of cystine into the cell. So far, there are no data on the glutathione levels in MoCD or iSOD patients available, but previous in vitro studies have shown a rapid decrease in the levels of glutathione disulfide, followed by a slow decrease in glutathione levels in rat hepatocytes upon treatment with high concentrations of sulfite [118]. Moreover, glutamate excitotoxicity, which presents as mechanistically similar to SSC-induced toxicity, has also been shown to induce a significant decrease in intracellular glutathione levels in the brain [119]. Based on these findings, it may be hypothesized that ferroptosis contributes to the pathophysiology of MoCD and iSOD, in particular to the rapidly progressing cell death, thus rendering ferroptosis inhibitors a promising treatment strategy to be investigated in the future.

### 4.8. Is H_2_S Involved in the Pathophysiology of MoCD?

A hallmark in the diagnosis of MoCD and SOXD is the accumulation and excretion of thiosulfate, a catabolic product of H_2_S metabolism. There is only one other disorder known to result in massive thiosulfate accumulation, known as ethylmalonic encephalopathy. Mutations causing a defect in the enzyme ethylmalonic encephalopathy protein 1 (ETHE1), also known as persulfide dioxygenase, result in the accumulation of H_2_S and the loss of clearance of small-molecule persulfides such as GSSH. As a consequence, TST-dependent oxidation of persulfides becomes the only path to catabolize H_2_S-products, thus leading to elevated levels of thiosulfate. Therefore, thiosulfate accumulation in MoCD and iSOD hints at a re-routing of sulfur flux into the H_2_S pathway, leading to increased H_2_S turnover. To which extent thiosulfate accumulation will indicate the increased catabolism and/or increased biosynthesis of H_2_S requires further investigation. 

A hallmark of H_2_S-based signaling is the formation of S-persulfidated species in small molecules (cystine-persulfide, glutathione-persulfide, etc.) as well as in protein cysteine side chains. Levels of protein S-persulfidation have been recently associated with cellular stress resistance and longevity in various model systems [120]. On the other hand, elevated H_2_S leads to the inhibition of mitochondrial respiration by blocking cytochrome c oxidase [121,122]. In addition, a recent study demonstrated that in H_2_S-inhibited mitochondria, accumulating H_2_S is further metabolized by SQOR, leading to the ‘reversal’ of the citrate cycle, glutaminolysis, and lipogenesis [123].

In aggregate, accumulating thiosulfate in MoCD and iSOD gives rise to the proposal that alterations in H_2_S homeostasis may contribute to a larger extent to mitochondrial dysfunction than initially anticipated, and recent data support this notion [124,125], being in line with other data showing sulfite-dependent changes in mitochondrial morphology [126].

A recent study in Moco-deficient *Caenorhabditis elegans* demonstrated that the lethal symptoms of MoCD in worms can be suppressed by two principles [127]: (i) On one hand, the uptake of Moco by feeding worms with Moco containing *E. coli* or providing an in vitro source of protein-bound Moco [128] was sufficient to protect the worms from sulfite toxicity. This finding suggests that not only cPMP but also Moco may be used to treat MoCD in the future. (ii) Second, a genetic suppressor screen identified CDO and CSE as potential targets for upstream inhibition to reduce cysteine catabolism and cysteine de novo synthesis, respectively. Again, those promising findings need to be verified in more disease-relevant preclinical models.

## 5. Future Perspectives

MoCD and iSOD have been known for nearly five decades, and major progress in understanding the biosynthesis of Moco, the function of individual gene products, and the contribution of Mo-enzymes to the disease pathology has been made. With the first therapy of MoCD type A being recently approved by the FDA, the number of treated patients will grow. Most important will be a timely diagnosis of patients to avoid irreversible brain damage due to sulfite intoxication. Methods such as newborn screening as well as targeted genetic testing will be instrumental in achieving this goal. Further understanding of the underlying disease mechanism, in particular, the role of H_2_S, will not only improve therapeutic outcomes but it may also open Moco enzyme research to other fields and challenges in medicine. For example, the benefits of dietary restriction have been recently demonstrated to involve the metabolism of sulfur-containing amino acids. We have recently demonstrated that stress-induced acute kidney injury can be prevented by a reduction of methionine and cysteine in the diet [129].

Therapies for MoCD type B, type C, and iSOD are still lacking; here, we have discussed biochemical principles that require experimental testing in preclinical models of the respective disorders. One additional treatment strategy holding great potential is gene therapy, which, in the past, has already been used for the successful treatment of metabolic disorders such as hemophilia B [130], lipoprotein lipase deficiency [131], and metachromatic leukodystrophy [132]. The most promising and widely used approach involves the usage of virus-mediated gene additions. It aims to rescue the phenotype of genetic diseases by either introducing a novel gene that helps to fight the consequences of a disease or by restoring the function of a defective gene by inserting a healthy copy [133]. As MoCD and iSOD are both single-gene disorders, they would be particularly suitable targets for this type of therapy.

## Figures and Tables

**Figure 1 molecules-27-06896-f001:**
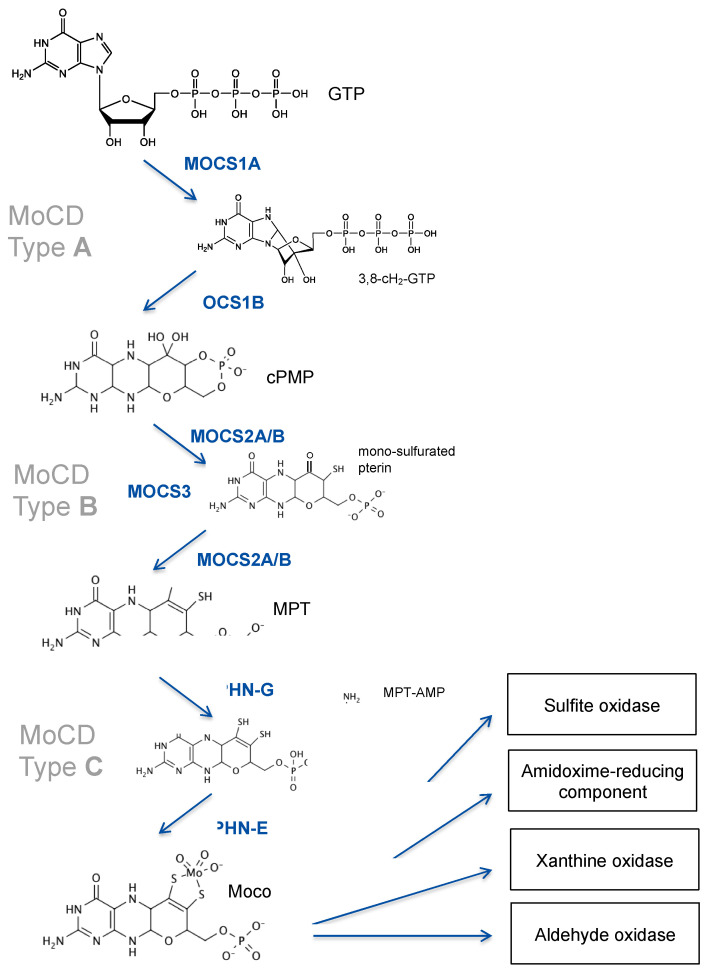
**Moco biosynthesis; Mo-enzymes and deficiencies.** Intermediates of Moco synthesis are 3′ 8-cyclo 7,8-dihydro-GTP, cyclic pyranopterin monophosphate (cPMP), mono-sulfurate pterin, metal binding pterin (MPT), and adenylated MPT (MPT-AMP). Proteins and domains (GPHN) involved in Moco synthesis are shown. MOCS3 is involved in both steps of cPMP-to-MPT conversion as it is required for the thiolation of MOCS2A. Deficiencies in MOCS1, MOCS2/3, and GPHN cause MoCD type A, type B, and type C, respectively.

**Figure 2 molecules-27-06896-f002:**
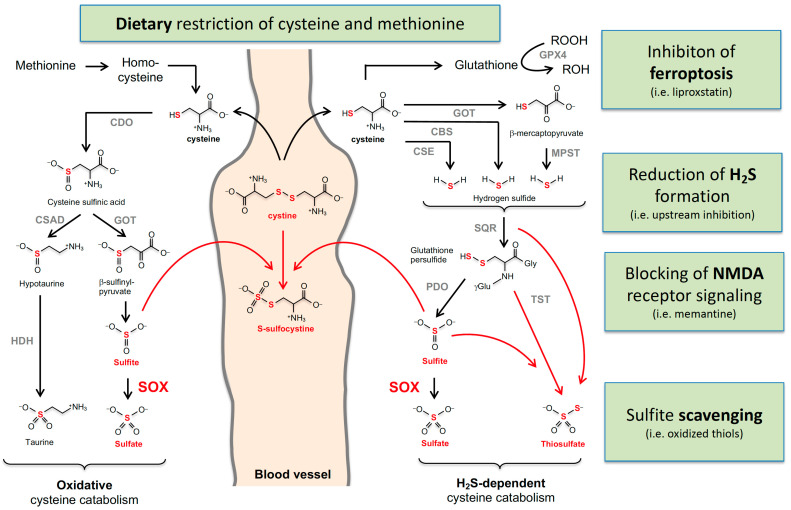
**Cysteine catabolism and sulfur-containing metabolites that are altered in MoCD and SOX deficiency.** Oxidative and H_2_S-dependent cysteine catabolisms are summarized with the involved enzymes and metabolites. Accumulating metabolites in MoCD and SO deficiency are labeled in red. Cystine and homocysteine were found to be reduced. The arrows from methionine to homocysteine, cysteine, and glutathione represent multiple enzymatic reactions. Enzyme abbreviations are in alphabetical order: GOT, glutamate oxaloacetate transaminase; CBS, cystathionine β-synthase; CSE, cystathionine γ-lyase; CDO, cysteine dioxygenase; CSD; cysteine sulfinate decarboxylase; HDH, hypoxanthine dehydrogenase; MPST, 3-mercaptopyruvate sulfurtransferase; PSD, persulfide dioxygenase; SO, sulfite oxidase; SQR, quinone oxidoreductase; TST, thiosulfate sulfur transferase.

**Table 1 molecules-27-06896-t001:** List of MoCD type B mutations.

Gene	Nucleotide Change	Amino Acid Change/ Predicted Effect	Onset	Reference
MOCS2A	c.-9_14del23	Initiation failure	Early	[50]
	c.-9G > C	-	Early	[51]
	c.1A > G	Initiation failure	-	[42]
	c.3G > A	p.M1I (Initiation failure)	Early	[52,53]
	c.16C > T (c.44C > T)	p.Q6X	Early	[54]
	c.19G > T (c.47G > T)	p.V7F *	Early	[54,55]
	c.33T > G	p.Y11X	-	[56]
	c.45T > A	p.S15R *	-	[57]
	c.88C > T	p.Q30X	-	[57]
	c.106C > T	p.Q36X	-	[57]
	c.130C > T	p.R44X	Early	[58]
MOCS2A/B	c.218T > C	p.L73P (MOCS2A)	Early	[59]
	c.220C > T	p.Q74X (MOCS2A)	Late	[42]
	c.226G > A	p.G76R (MOCS2A)	Early	[26]
	c.252insC	Premature termination	Early	[60]
	c.265T > C	p.X89Q (MOCS2A), silenced p.D26D (MOCS2B)	Late	[61]
	c.266A > G	p.X89W (MOCS2A), p.D26G (MOCS2B)	Late	[61]
MOCS2B	c.413G > A	p.G76R	Early	[57]
	c.419C > T	p.S140F *	Early	[62]
	c.493 T > C	p.W165R	Late	[63]
	c.501delA	p.K105fs	-	[42]
	c.501 + 2delT	Disruption of splice site	Early	[62]
	c.533_536delGTCA	p.V116fs(Premature termination)	-	[53]
	c.564G > C	p.G126A	-	[57,64]
	c.564 + 1G > A	Skipping exon 5	Early	[42]
	c.635_637delGCT	p.A150del *	Early	[57]
	c.658_664delTTTAAAAinsG	p.L158_K159del	-	[57]
	c.689G > A	p.E168K *	-	[57]
	c.714_718delGGAAA	p.G178fs (Premature termination)	-	[57]
	c.726_727delAA	p.K180fs (Premature termination)	Late	[57,64]
	c.754A > C	p.X189Y	-	[56]
MOCS3	c.325C > G	p.L109V	Early	[65]
	c.769G > A	p.A257T	Late	[66]
	c.1375C > T	p.Q459X	Early	[65]

*: Variants that have been characterized. -: Not mentioned in the case report.

## Data Availability

Not applicable.

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
