# Peer review of "Molybdenum Cofactor Deficiency in Humans"

_molecules, 2022, doi:10.3390/molecules27206896_

Round 1
Reviewer 1 Report
This contribution has reviewed three types of Moco deficiency (MoCD) based on the origins of the Moco biosynthesis step impaired. Majority of MoCD are found to be Type A MoCD which fortunately can be treated with a substitution therapy of chemically synthesized and FDA-approved cPMP. However, Type B and C MoCD are more complicated and so far not curable. Many patient cases including their disease onset, severity, treatments, and results are introduced. It is informative. Recommend publishing in Molecules with minor changes.
1. Title, “Molybdenum cofactor deficiency in men”. Suggest changing "men" to “human”.
2 2. Line 68, add reference https://doi.org/10.1038/s41557-021-00714-1 after “MPT-AMP”;
3 3. Line 289, SOXD changes to iSOD;
4 4. Line 302, add “(” to “Figure 2)”;
5 5. Line 307, delete one “three”;
6 6. Line 338, typo “aujthorities”;
7 7. Line 345 typo “paediatrician”;
8 8. Line 356, “the chemical synthesis of cPMP has been achieved in 20xx [71]”, which year?
9 9. Line 359, “clinical phase II/III study (REF).”, please add the reference;
1 10. Line 381, “In a recent study by Abe et al. (2021),” add reference here;
1 11. Line 536, 542, and 545, H2S should be corrected as H2S;
1 12. Line 539, add reference.
Author Response
Dear Dr. Prof. Mendel,
Dear Ralf,
We thank you and the referees for the fast and positive response to our submitted manuscript on “Molybdenum cofactor deficiency in human” (changed title as suggested by both reviewers) as review article in Molecules.
All suggested minor changes have been corrected as suggested by the referees.
In addition, we have found some other typos and missing or non-formatted
references that has been corrected in the revised version. Furthermore, we
added an additional short paragraph on the molybdate treatment in iSOX and
MoCD. For comparison we also include a tracked-change-version of the revised
manuscript.
Please do not hesitate to contact me (Tel: +49 221 470 6441, Fax: +49 221 470
5092, gschwarz@uni-koeln.de) for any additional information you might need.
We look forward seeing this review published in Molecules
Sincerely,
Guenter Schwarz
*****************************
Poin-by-point response:
Reviewer #1:
Answer: All minor changes, typographical errors and missing references have been corrected and/or added.
- Title, “Molybdenum cofactor deficiency in men”. Suggest changing "men" to “human”.
- Line 68, add reference https://doi.org/10.1038/s41557-021-00714-1 after “MPT-AMP”;
- Line 289, SOXD changes to iSOD;
- Line 302, add “(” to “Figure 2)”;
- Line 307, delete one “three”;
- Line 338, typo “aujthorities”;
- Line 345 typo “paediatrician”;
- Line 356, “the chemical synthesis of cPMP has been achieved in 20xx [71]”, which year?
- Line 359, “clinical phase II/III study (REF).”, please add the reference;
- Line 381, “In a recent study by Abe et al. (2021),” add reference here;
- Line 536, 542, and 545, H2S should be corrected as H2S;
- Line 539, add reference.
Reviewer #2
Answer: All minor changes, typographical errors and missing references have been corrected and/or added. Some of which were overlapping with reviewer #1
Firstly, “men” in the title and abstract is misleading. Certainly women are affected as well. I suggest using “humans”, “human beings”, or “humankind” instead.
In the abstract, define “cystine” so that readers will not consider this a typo (cysteine).
Line 40 No comma before “that”
Line 72 Define gephyrin (no clear this a protein)
Line 96 Although common, I suggest defining the abbreviation “EEG”
Line 122 Add “enzymes” behind “(SAM)”
Line 127 Add “clusters” behind “[4Fe-4S]”
Line 164 Delete “Without a doubt”
Line 165 Add “in particular” behind “MoCD type A”
Line 176 Add “an” before “unusual”
Line 204 Do not abbreviate “Approx.”
Line 219 Delete gap in “50_%”
Line 223 Italicize “in vitro”, add period after ref. [50]
Line 228 Delete period before ref. [15]
Line 241 Delete period after refs. [53,54]
Line 286 “highly related” is weird – “directly related”?
Line 333 “the” is written twice
Line 356 Correct “20xx”
Line 365 Delete “/”
Line 399 I think “and” should be “or”
Line 423 Replace “Unfortunately” by “However” (emotional)
In Section 4.7, “H2S” should be “H2S” (several times)
Line 539 Add reference
Line 550 Italicize “in vitro”
Line 553 Italicize “de novo”
Line 559 “have” should be “has” (relates to “progress”)
Lastly, I recommend an additional table listing the huge number of discussed proteins/enzymes and their respective abbreviation.

Reviewer 2 Report
The review article „Molybenum Cofactor Deficieny in Men“ by Johannes, Fu, and Schwarz is sufficiently comprehensive, well-written, and timely. It will be a valuable addition to the topical focus.
I recommend publication after very minor revisions. Please find a number of highlighted typos and suggestions below.
Firstly, “men” in the title and abstract is misleading. Certainly women are affected as well. I suggest using “humans”, “human beings”, or “humankind” instead.
In the abstract, define “cystine” so that readers will not consider this a typo (cysteine).
Line 40 No comma before “that”
Line 72 Define gephyrin (no clear this a protein)
Line 96 Although common, I suggest defining the abbreviation “EEG”
Line 122 Add “enzymes” behind “(SAM)”
Line 127 Add “clusters” behind “[4Fe-4S]”
Line 164 Delete “Without a doubt”
Line 165 Add “in particular” behind “MoCD type A”
Line 176 Add “an” before “unusual”
Line 204 Do not abbreviate “Approx.”
Line 219 Delete gap in “50_%”
Line 223 Italicize “in vitro”, add period after ref. [50]
Line 228 Delete period before ref. [15]
Line 241 Delete period after refs. [53,54]
Line 286 “highly related” is weird – “directly related”?
Line 333 “the” is written twice
Line 356 Correct “20xx”
Line 365 Delete “/”
Line 399 I think “and” should be “or”
Line 423 Replace “Unfortunately” by “However” (emotional)
In Section 4.7, “H2S” should be “H2S” (several times)
Line 539 Add reference
Line 550 Italicize “in vitro”
Line 553 Italicize “de novo”
Line 559 “have” should be “has” (relates to “progress”)
Lastly, I recommend an additional table listing the huge number of discussed proteins/enzymes and their respective abbreviation.
Author Response
Dear Dr. Prof. Mendel,
Dear Ralf,
We thank you and the referees for the fast and positive response to our submitted manuscript on “Molybdenum cofactor deficiency in human” (changed title as suggested by both reviewers) as review article in Molecules.
All suggested minor changes have been corrected as suggested by the referees.
In addition, we have found some other typos and missing or non-formatted
references that has been corrected in the revised version. Furthermore, we
added an additional short paragraph on the molybdate treatment in iSOX and
MoCD. For comparison we also include a tracked-change-version of the revised
manuscript.
Please do not hesitate to contact me (Tel: +49 221 470 6441, Fax: +49 221 470
5092, gschwarz@uni-koeln.de) for any additional information you might need.
We look forward seeing this review published in Molecules
Sincerely,
Guenter Schwarz
*****************************
Reviewer #1:
Answer: All minor changes, typographical errors and missing references have been corrected and/or added.
- Title, “Molybdenum cofactor deficiency in men”. Suggest changing "men" to “human”.
- Line 68, add reference https://doi.org/10.1038/s41557-021-00714-1 after “MPT-AMP”;
- Line 289, SOXD changes to iSOD;
- Line 302, add “(” to “Figure 2)”;
- Line 307, delete one “three”;
- Line 338, typo “aujthorities”;
- Line 345 typo “paediatrician”;
- Line 356, “the chemical synthesis of cPMP has been achieved in 20xx [71]”, which year?
- Line 359, “clinical phase II/III study (REF).”, please add the reference;
- Line 381, “In a recent study by Abe et al. (2021),” add reference here;
- Line 536, 542, and 545, H2S should be corrected as H2S;
- Line 539, add reference.
Reviewer #2
Answer: All minor changes, typographical errors and missing references have been corrected and/or added. Some of which were overlapping with reviewer #1
Firstly, “men” in the title and abstract is misleading. Certainly women are affected as well. I suggest using “humans”, “human beings”, or “humankind” instead.
In the abstract, define “cystine” so that readers will not consider this a typo (cysteine).
Line 40 No comma before “that”
Line 72 Define gephyrin (no clear this a protein)
Line 96 Although common, I suggest defining the abbreviation “EEG”
Line 122 Add “enzymes” behind “(SAM)”
Line 127 Add “clusters” behind “[4Fe-4S]”
Line 164 Delete “Without a doubt”
Line 165 Add “in particular” behind “MoCD type A”
Line 176 Add “an” before “unusual”
Line 204 Do not abbreviate “Approx.”
Line 219 Delete gap in “50_%”
Line 223 Italicize “in vitro”, add period after ref. [50]
Line 228 Delete period before ref. [15]
Line 241 Delete period after refs. [53,54]
Line 286 “highly related” is weird – “directly related”?
Line 333 “the” is written twice
Line 356 Correct “20xx”
Line 365 Delete “/”
Line 399 I think “and” should be “or”
Line 423 Replace “Unfortunately” by “However” (emotional)
In Section 4.7, “H2S” should be “H2S” (several times)
Line 539 Add reference
Line 550 Italicize “in vitro”
Line 553 Italicize “de novo”
Line 559 “have” should be “has” (relates to “progress”)
Lastly, I recommend an additional table listing the huge number of discussed proteins/enzymes and their respective abbreviation.